# Mathematical Modeling and Analysis of the Dynamics of RNA Viruses in Presence of Immunity and Treatment: A Case Study of SARS-CoV-2

**DOI:** 10.3390/vaccines11020201

**Published:** 2023-01-17

**Authors:** Khalid Hattaf, Mly Ismail El Karimi, Ahmed A. Mohsen, Zakaria Hajhouji, Majda El Younoussi, Noura Yousfi

**Affiliations:** 1Equipe de Recherche en Modélisation et Enseignement des Mathématiques (ERMEM), Centre Régional des Métiers de l’Education et de la Formation (CRMEF), Derb Ghalef, Casablanca 20340, Morocco; 2Laboratory of Analysis, Modeling and Simulation (LAMS), Faculty of Sciences Ben M’Sick, Hassan II University of Casablanca, Sidi Othman, Casablanca P.O. Box 7955, Morocco; 3Department of Mathematics, College of Education for Pure Science (Ibn Al-Haitham), University of Baghdad, Baghdad 10071, Iraq; 4Ministry of Education, Baghdad 10071, Iraq

**Keywords:** RNA viruses, SARS-CoV-2, immunity, antiviral treatment, mathematical modeling

## Abstract

The emergence of novel RNA viruses like SARS-CoV-2 poses a greater threat to human health. Thus, the main objective of this article is to develop a new mathematical model with a view to better understand the evolutionary behavior of such viruses inside the human body and to determine control strategies to deal with this type of threat. The developed model takes into account two modes of transmission and both classes of infected cells that are latently infected cells and actively infected cells that produce virus particles. The cure of infected cells in latent period as well as the lytic and non-lytic immune response are considered into the model. We first show that the developed model is well-posed from the biological point of view by proving the non-negativity and boundedness of model’s solutions. Our analytical results show that the dynamical behavior of the model is fully determined by two threshold parameters one for viral infection and the other for humoral immunity. The effect of antiviral treatment is also investigated. Furthermore, numerical simulations are presented in order to illustrate our analytical results.

## 1. Introduction

RNA viruses are one of the main causes of human infectious diseases and continue to be a major public health problem worldwide, especially after the emergence of new types of such viruses. For instance, coronavirus disease 2019 (COVID-19) is an infectious disease caused by a novel coronavirus named severe acute respiratory syndrome coronavirus 2 (SARS-CoV-2) by the International Committee on Taxonomy of Viruses (ICTV) [1]. Since the first case emerged in Wuhan, China at the end of 2019, the disease has spread rapidly from country to country, causing enormous economic damage and many deaths around the world. According to World Health Organization (WHO) statistics as 18 September 2022 [2], SARS-CoV-2 has infected more than 609 million people worldwide and caused more than 6.5 million deaths.

Despite compliance with preventive measures and the administration of vaccines in several countries, SARS-CoV-2 keeps on spreading. This has led researchers to develop treatments or vaccines to prevent or stop the progression of this disease. Antiviral treatment for COVID-19 remains a challenge. Some drugs have shown promising antiviral activity. This is the case of Paxlovid which has been shown to be highly effective against SARS-CoV-2 with an effectiveness rate of 89% in high-risk patients [3].

Recently, modeling the propagation of SARS-CoV-2 in human population has attracted the attention of several researchers [4,5,6,7,8]. However, there are few models that describe the dynamics of this virus inside the human body in presence of immunity. For this reason, Hattaf and Yousfi [9] proposed a mathematical within-host model describing the interactions between SARS-CoV-2, host pulmonary epithelial cells and cytotoxic T lymphocyte (CTL) cells. Chatterjee et al. [10] studied a SARS-CoV-2 infection model with antiviral drug and CTL immune response incorporating lytic and non-lytic immune responses effect. Another model for SARS-CoV-2 infection with CTL immune response and treatment was proposed in [11]. Following a SARS-CoV-2 infection, the humoral immune system triggers production of a variety of antibodies. IgM, IgA and IgG are the most frequently described antibodies involved in SARS-CoV-2 infection. IgM are the first to be activated, they are detected on day 4 [12] and they last for 20 days to a month before they disappear gradually [13]. IgA are produced in serum, saliva and bronchial mucosa. Serum IgA are detected on day 6 to 8 [14], they reach the maximum on day 20 to 22 [14,15]. High levels of serum IgA are correlated positively with severity of SARS-CoV-2 infection. Whereas mild disease is associated with temporary, late or even absent S-protein specific serum IgA [16]. This suggests that in mild cases mucosal of IgA are activate instead of systemic production. IgG are produced on day 10 to 14 [16,17], they reach the maximum on day 25 and they last for weeks [17]. Neutralizing activity is achieved by synergic effect of the three immunoglobulines classes [18]. Furthermore, humoral immunity exerted by antibodies to neutralize viral particles is more effective than cellular immunity mediated by CTL cells in some infections [19]. Motivated by the above mathematical and biological considerations, we will introduce a model that takes into account the role of antibody immune response in SARS-CoV-2 infection and other biological factors.

The main objective of this study is to answer the following research problem: how can we describe the dynamics of SARS-CoV-2 infection in the presence of humoral immunity and antiviral treatment? And what is the impact of this treatment on this dynamics? To do this, we will propose a mathematical model that incorporates the two fundamental modes of transmission of the virus which are direct cell-to-cell transmission via virological synapses and virus-to-cell infection via the extracellular environments, and takes into account the effects of cure of latently infected cells, lytic and non-lytic humoral immune response. In addition, we study the impact of antiviral treatment on the dynamics of SARS-CoV-2 infection via numerical simulations.

## 2. Materials and Methods

### 2.1. Mathematical Formulation

We use a new mathematical model to describe the dynamics of RNA viruses such as SARS-CoV-2 that incorporates two modes of transmission (virus-to-cell and cell-to-cell), two classes of infected cells, humoral immunity and antiviral treatment. The model is formulated by the following nonlinear system of ordinary differential equations (ODEs):(1)$$\left\{\begin{array}{ccc} \frac{dS}{dt} & = & \sigma -\mu_{1}S-\frac{\beta_{1}SV}{1+q_{1}W}-\frac{\beta_{2}SI}{1+q_{2}W}+\rho L,\\ \frac{dL}{dt} & = & \frac{\beta_{1}SV}{1+q_{1}W}+\frac{\beta_{2}SI}{1+q_{2}W}-\left(\mu_{2}+\delta +\rho \right)L,\\ \frac{dI}{dt} & = & \delta L-\mu_{3}I,\\ \frac{dV}{dt} & = & k\left(1-\epsilon \right)I-\mu_{4}V-pVW,\\ \frac{dW}{dt} & = & rVW-\mu_{5}W, \end{array}\right.$$
where the uninfected cells (S) are generated at rate σ, die at rate $\mu_{1}S$ and become infected either by free virus particles at rate $\beta_{1}SV$ or by direct contact with infected cells at rate $\beta_{2}SI$. The two modes of transmission are inhibited by non-lytic humoral immune response at rate $1+q_{1}W$ and $1+q_{2}W$, respectively. The latently infected cells (L) die at rate $\mu_{2}L$ and become productively infected cells rate δL. Also, the latently infected cells are assumed to be cured at rate ρL, resulting from the clearance of virus through the non-cytolytic process as for HCV infection in [20] and HIV in [21,22]. The cure of infected epithelial cells was also considered in a recent work of SARS-CoV-2 [23]. The productively infected cells (I) die at rate $\mu_{3}I$. Free viruses (V) are produced by infected cells at rate kI, cleared at rate $\mu_{4}V$ and neutralized by antibodies at rate pVW. Antibodies develop in response to free virus at rate rVW and decay at rate $\mu_{5}W$. Here, the parameter ϵ represents the effectiveness of the antiviral treatment which blocks the production of viral particles. The flow diagram of the model is shown in Figure 1.

Most viruses can spread via two modes: by virus-to-cell infection and through direct cell-cell contact [24,25,26]. A recent study provided evidence that SARS-CoV-2 spreads through cell-cell contact in cultures, mediated by the spike glycoprotein [27]. Furthermore, it is known that antibodies neutralize free virus particles and inhibit the infection of susceptible cells [28]. They also contribute significantly to non-lytic antiviral activity [29]. For this reason, both modes of transmission with the lytic and non-lytic immune response are considered into the model.

On the other hand, it is very important to note that the SARS-CoV-2 model presented by system (Equation 1) includes many mathematical models for viral infection existing in the literature. For instance, we get the model of Rong et al. [21] when $q_{1}=0$, $\beta_{2}=0$ and both treatment and humoral immunity are ignored. The global stability of the model [21] was investigated in [30]. In addition, the model of Baccam et al. [31] is a special case of system (Equation 1), it suffices to neglect immunity and take σ=0, $\mu_{1}=\mu_{2}=\rho =0$, $\beta_{2}=0$ and ϵ=0. The last model presented in [31] was recently used by Rodriguez and Dobrovolny [32] to determine viral kinetics parameters for young and aged SARS-CoV-2 infected macaques. In the case where latently infected cells not revert back to susceptible and when antibodies do not reduce cell-to-cell transmission, we have ρ=0, $q_{2}=0$ and system (Equation 1) reduces to the following model:(2)$$\left\{\begin{array}{ccc} \frac{dS}{dt} & = & \sigma -\mu_{1}S-\frac{\beta_{1}SV}{1+q_{1}W}-\beta_{2}SI,\\ \frac{dL}{dt} & = & \frac{\beta_{1}SV}{1+q_{1}W}+\beta_{2}SI-\left(\mu_{2}+\delta \right)L,\\ \frac{dI}{dt} & = & \delta L-\mu_{3}I,\\ \frac{dV}{dt} & = & k\left(1-\epsilon \right)I-\mu_{4}V-pVW,\\ \frac{dW}{dt} & = & rVW-\mu_{5}W, \end{array}\right.$$

### 2.2. Equilibria and Threshold Parameters

The equilibria of the model are the stationary solutions of system (Equation 1). Then they satisfied the following equations: (3)$$\begin{array}{ccc} \sigma -\mu_{1}S-\frac{\beta_{1}SV}{1+q_{1}W}-\frac{\beta_{2}SI}{1+q_{2}W}+\rho L & = & 0, \end{array}$$(4)$$\begin{array}{ccc} \frac{\beta_{1}SV}{1+q_{1}W}+\frac{\beta_{2}SI}{1+q_{2}W}-\left(\mu_{2}+\delta +\rho \right)L & = & 0, \end{array}$$(5)$$\begin{array}{ccc} \;\;\;\;\;\;\;\;\;\;\;\delta L-\mu_{3}I & = & 0, \end{array}$$(6)$$\begin{array}{ccc} \;\;\;\;\;k\left(1-\epsilon \right)I-\mu_{4}V-pVW & = & 0, \end{array}$$(7)$$\begin{array}{ccc} \;\;\;\;\;\;\;\;\;\;rVW-\mu_{5}W & = & 0. \end{array}$$
From Equation (Equation 7), we have W=0 or $V=\frac{\mu_{5}}{r}$. So, we discuss two cases.

If W=0, then $L=\frac{\mu_{3}I}{\delta }$, $V=\frac{k(1-\epsilon )I}{\mu_{4}}$ and
$$I\left(\frac{k\left(1-\epsilon \right)\beta_{1}S}{\mu_{4}}+\beta_{2}S-\frac{\left(\mu_{2}+\delta +\rho \right)\mu_{3}}{\delta }\right)=0.$$ **(i)** When I=0, we have L=V=0 and according to (Equation 3) we get $S=\frac{\sigma }{\mu_{1}}$. Thus, model (Equation 1) admits an equilibrium point of the form $E_{0}=\left(\frac{\sigma }{\mu_{1}},0,0,0,0\right)$. This point is called the infection-free equilibrium which corresponding to the healthy state of the patient. **(ii)** When I≠0, we have $S=\frac{\mu_{3}\mu_{4}\left(\mu_{2}+\delta +\rho \right)}{\delta k\left(1-\epsilon \right)\beta_{1}+\delta \mu_{4}\beta_{2}}$. By summing (Equation 3) and (Equation 4), we obtain $L=\frac{\sigma -\mu_{1}S}{\mu_{2}+\delta }=\frac{\delta \sigma \left[k\left(1-\epsilon \right)\beta_{1}+\mu_{4}\beta_{2}\right]-\mu_{1}\mu_{3}\mu_{4}\left(\mu_{2}+\delta +\rho \right)}{\delta \left(\mu_{2}+\delta \right)\left[k\left(1-\epsilon \right)\beta_{1}+\mu_{4}\beta_{2}\right]}$. Since I>0, we have $\delta \sigma \left[k\left(1-\epsilon \right)\beta_{1}+\mu_{4}\beta_{2}\right]>\mu_{1}\mu_{3}\mu_{4}\left(\mu_{2}+\delta +\rho \right)$. This leads to $R_{0}>1$, where
(8)$$R_{0}=\frac{\sigma \delta [k\left(1-\epsilon \right)\beta_{1}+\mu_{4}\beta_{2}]}{\mu_{1}\mu_{3}\mu_{4}\left(\mu_{2}+\delta +\rho \right)}.$$The threshold parameter $R_{0}$ is called the basic reproduction number. Biologically, this threshold parameter represents the average number of secondary infections produced by one productively infected cell at the beginning of infection. It can be rewritten as $R_{01}+R_{02}$, where $R_{01}=\frac{k\delta \sigma \beta_{1}\left(1-\epsilon \right)}{\mu_{1}\mu_{2}\mu_{4}\left(\mu_{2}+\delta +\rho \right)}$ is the basic reproduction number of the virus-to-cell transmission mode and $R_{02}=\frac{\sigma \delta \beta_{2}}{\mu_{1}\mu_{3}\left(\mu_{2}+\delta +\rho \right)}$ is the basic reproduction number of the cell-to-cell transmission mode.When $R_{0}>1$, model (Equation 1) has another biological equilibrium called the infection equilibrium without humoral immunity of the form $E_{1}=\left(S_{1},L_{1},I_{1},V_{1},0\right)$, where $S_{1}=\frac{\sigma }{\mu_{1}R_{0}}$, $L_{1}=\frac{\sigma (R_{0}-1)}{\left(\mu_{2}+\delta \right)R_{0}}$, $I_{1}=\frac{\delta \sigma (R_{0}-1)}{\mu_{3}\left(\mu_{2}+\delta \right)R_{0}}$ and $V_{1}=\frac{k\delta \sigma \left(1-\epsilon \right)\left(R_{0}-1\right)}{\mu_{3}\mu_{4}\left(\mu_{2}+\delta \right)R_{0}}$.If W≠0, then $V=\frac{\mu_{5}}{r}$. It follows from (Equation 3) to (Equation 6) that $L=\frac{\sigma -\mu_{1}S}{\mu_{2}+\delta }$, $I=\frac{\delta (\sigma -\mu_{1}S)}{\mu_{3}\left(\mu_{2}+\delta \right)}$, $W=\frac{rk\delta \left(1-\epsilon \right)\left(\sigma -\mu_{1}S\right)-\mu_{3}\mu_{4}\mu_{5}\left(\mu_{2}+\delta \right)}{p\mu_{3}\mu_{5}\left(\mu_{2}+\delta \right)}$ and
$$\frac{\mu_{5}\left(\mu_{2}+\delta \right)\beta_{1}S}{r(1+q_{1}W)}+\frac{\delta \beta_{2}S\left(\sigma -\mu_{1}S\right)}{\mu_{3}\left(1+q_{2}W\right)}=\left(\mu_{2}+\delta +\rho \right)\left(\sigma -\mu_{1}S\right).$$Since W≥0, we have $S\leq \frac{\sigma }{\mu_{1}}-\frac{\mu_{3}\mu_{4}\mu_{5}\left(\mu_{2}+\delta \right)}{rk\delta \mu_{1}\left(1-\epsilon \right)}$. This implies that there is no biological equilibrium when $S>\frac{\sigma }{\mu_{1}}-\frac{\mu_{3}\mu_{4}\mu_{5}\left(\mu_{2}+\delta \right)}{rk\delta \mu_{1}\left(1-\epsilon \right)}$ or $\frac{\sigma }{\mu_{1}}-\frac{\mu_{3}\mu_{4}\mu_{5}\left(\mu_{2}+\delta \right)}{rk\delta \mu_{1}\left(1-\epsilon \right)}\leq 0$. Let $s^{*}=\frac{\sigma }{\mu_{1}}-\frac{\mu_{3}\mu_{4}\mu_{5}\left(\mu_{2}+\delta \right)}{rk\delta \mu_{1}\left(1-\epsilon \right)}$ and F be the function defined on the closed interval $\left[0,s^{*}\right]$ as follows
$$\begin{array}{ccc} F(S) & = & \frac{\mu_{5}\left(\mu_{2}+\delta \right)\beta_{1}S}{r(1+q_{1}g\left(S\right))}+\frac{\delta \beta_{2}S\left(\sigma -\mu_{1}S\right)}{\mu_{3}\left(1+q_{2}g\left(S\right)\right)}-\left(\mu_{2}+\delta +\rho \right)\left(\sigma -\mu_{1}S\right), \end{array}$$
where $g\left(S\right)=\frac{rk\delta \left(1-\epsilon \right)\left(\sigma -\mu_{1}S\right)-\mu_{3}\mu_{4}\mu_{5}\left(\mu_{2}+\delta \right)}{p\mu_{3}\mu_{5}\left(\mu_{2}+\delta \right)}$. On the other hand, we have $F\left(0\right)=-\sigma \left(\mu_{2}+\delta +\rho \right)<0$ and
$$\begin{array}{ccc} F^{\prime }\left(S\right) & = & \frac{\mu_{5}\left(\mu_{2}+\delta \right)\beta_{1}\left(1+q_{1}g\left(S\right)-q_{1}Sg^{\prime }\left(S\right)\right)}{r{\left(1+q_{1}g\left(S\right)\right)}^{2}}\\ & & +\frac{\delta \beta_{2}\left(\sigma -\mu_{1}S\right)\left(1+q_{2}g\left(S\right)-q_{2}Sg^{\prime }\left(S\right)\right)}{\mu_{3}{\left(1+q_{2}g\left(S\right)\right)}^{2}}\\ & & +\mu_{1}\left(\mu_{2}+\delta +\rho -\frac{\delta \beta_{2}S}{\mu_{3}\left(1+q_{2}g\left(S\right)\right)}\right). \end{array}$$When the humoral immune response has not been established, we have $rV_{1}-\mu_{5}\leq 0$. Hence, we define another threshold parameter called the reproduction number for humoral immunity as follows
(9)$$R_{1}^{W}=\frac{rV_{1}}{\mu_{5}},$$
where $\frac{1}{\mu_{5}}$ is the average life span of antibodies and $V_{1}$ is the quantity of viruses at the steady state $E_{1}$. So, the number $R_{1}^{W}$ can biologically determine the average number of antibodies activated by viral particles.As $F\left(s^{*}\right)=\frac{\mu_{3}\mu_{4}\mu_{5}^{2}{\left(\mu_{2}+\delta \right)}^{2}\left[k\left(1-\epsilon \right)\beta_{1}+\mu_{4}\beta_{2}\right]}{\delta \mu_{1}r^{2}k^{2}{\left(1-\epsilon \right)}^{2}}\left(R_{1}^{W}-1\right)>0$ if $R_{1}^{W}>1$, we deduce that there exists a $S_{2}\in \left(0,s^{*}\right)$ such that $F(S_{2})=0$. Further, we have $F^{\prime }\left(S_{2}\right)>0$. This establishes the uniqueness of $S_{2}$ and therefore model (Equation 1) has an unique infection equilibrium point with humoral immunity $E_{2}=\left(S_{2},L_{2},I_{2},V_{2},W_{2}\right)$ when $R_{1}^{W}>1$, where $S_{2}\in \left(0,s^{*}\right)$, $L_{2}=\frac{\sigma -\mu_{1}S_{2}}{\mu_{2}+\delta }$ $I_{2}=\frac{\delta (\sigma -\mu_{1}S_{2})}{\mu_{3}\left(\mu_{2}+\delta \right)}$, $V_{2}=\frac{\mu_{5}}{r}$ and $W_{2}=\frac{rk\delta \left(1-\epsilon \right)\left(\sigma -\mu_{1}S_{2}\right)-\mu_{3}\mu_{4}\mu_{5}\left(\mu_{2}+\delta \right)}{p\mu_{3}\mu_{5}\left(\mu_{2}+\delta \right)}$.

Summarizing the cases discussed above in the following result.

**Theorem** **1.**
 **(i)** 
*If $R_{0}\leq 1$, then model (Equation 1) has a unique infection-free equilibrium $E_{0}=\left(S_{0},0,0,0,0\right)$, where $S_{0}=\frac{\sigma }{\mu_{1}}$.*
 **(ii)** 
*If $R_{0}>1$, then model (Equation 1) has a unique infection equilibrium without humoral immunity $E_{1}=\left(S_{1},L_{1},I_{1},V_{1},0\right)$ besides $E_{0}$, where $S_{1}=\frac{\sigma }{\mu_{1}R_{0}}$, $L_{1}=\frac{\sigma (R_{0}-1)}{\left(\mu_{2}+\delta \right)R_{0}}$, $I_{1}=\frac{\delta \sigma (R_{0}-1)}{\mu_{3}\left(\mu_{2}+\delta \right)R_{0}}$ and $V_{1}=\frac{k\delta \sigma \left(1-\epsilon \right)\left(R_{0}-1\right)}{\mu_{3}\mu_{4}\left(\mu_{2}+\delta \right)R_{0}}$.*
 **(iii)** 
*If $R_{1}^{W}>1$, then model (Equation 1) has a unique infection equilibrium with humoral immunity $E_{2}=\left(S_{2},L_{2},I_{2},V_{2},W_{2}\right)$ besides $E_{0}$ and $E_{1}$, where*

*$S_{2}\in \left(0,\frac{\sigma }{\mu_{1}}-\frac{\mu_{3}\mu_{4}\mu_{5}\left(\mu_{2}+\delta \right)}{rk\delta \mu_{1}\left(1-\epsilon \right)}\right)$, $L_{2}=\frac{\sigma -\mu_{1}S_{2}}{\mu_{2}+\delta }$$I_{2}=\frac{\delta (\sigma -\mu_{1}S_{2})}{\mu_{3}\left(\mu_{2}+\delta \right)}$, $V_{2}=\frac{\mu_{5}}{r}$ and $W_{2}=\frac{rk\delta \left(1-\epsilon \right)\left(\sigma -\mu_{1}S_{2}\right)-\mu_{3}\mu_{4}\mu_{5}\left(\mu_{2}+\delta \right)}{p\mu_{3}\mu_{5}\left(\mu_{2}+\delta \right)}$.*



## 3. Analytical Results

This section is devoted to the analytical results including the well-posedness of the model and the stability analysis of equilibria.

### 3.1. Well-Posedness

The model presented by system (Equation 1) describes the dynamics of cells and virus which are nonnegative and bounded biological quantities. Since the second member of the system (Equation 1) is continuously differentiable, we deduce that the solution of (Equation 1) exists and it is unique. Next, we prove the non-negativity and boundedness of the solutions of (Equation 1) by establishing the following theorem.

**Theorem** **2.**
*Each solution of system (Equation 1) starting from nonnegative initial values remains nonnegative and bounded for all positive time.*


**Proof.** By system (Equation 1), we get
$$\begin{array}{ccc} \frac{dS}{dt}{|}_{S=0} & = & \sigma +\rho L>0\;\mathrm{for}\;\mathrm{all}\;L\geq 0,\\ \frac{dL}{dt}{|}_{L=0} & = & \frac{\beta_{1}SV}{1+q_{1}W}+\frac{\beta_{2}SI}{1+q_{2}W}\geq 0\;\mathrm{for}\;\mathrm{all}\;S,I,V,W\geq 0,\\ \frac{dI}{dt}{|}_{I=0} & = & \delta L\geq 0\;\mathrm{for}\;\mathrm{all}\;L\geq 0,\\ \frac{dV}{dt}{|}_{V=0} & = & k(1-\epsilon )I\geq 0\;\mathrm{for}\;\mathrm{all}\;I\geq 0,\\ \frac{dW}{dt}{|}_{W=0} & = & 0. \end{array}$$
According to Proposition B.7 of [33], we deduce that the solutions S(t), L(t), I(t), V(t) and W(t) are nonnegative if $\left(S\left(0\right),L\left(0\right),I\left(0\right),V\left(0\right),W\left(0\right)\right)\in R_{+}^{5}$.For the boundedness of solutions, we consider the following function
$$T\left(t\right)=S\left(t\right)+L\left(t\right)+I\left(t\right)+\frac{\mu_{3}}{2k(1-\epsilon )}V\left(t\right)+\frac{p\mu_{3}}{2k(1-\epsilon )r}W\left(t\right).$$
Then
$$\begin{array}{ccc} \frac{dT}{dt} & = & \sigma -\mu_{1}S\left(t\right)-\mu_{2}L\left(t\right)-\frac{\mu_{3}}{2}I\left(t\right)-\frac{\mu_{3}\mu_{4}}{2k(1-\epsilon )}V\left(t\right)-\frac{p\mu_{3}\mu_{5}}{2k(1-\epsilon )r}W\left(t\right)\\ & \leq & \sigma -\mu T(t), \end{array}$$
where $\mu =\min\{\mu_{1},\mu_{2},\frac{\mu_{3}}{2},\mu_{4},\mu_{5}\}$. Hence,
$$\underset{t\to \infty }{lim sup}T\left(t\right)\leq \frac{\sigma }{\mu }.$$
This ensures the boundedness of solutions. □

### 3.2. Stability Analysis

In this subsection, we analyze the dynamical behaviors of our model. First, we investigate the stability of the infection-free steady state $E_{0}$.

**Theorem** **3.**
*The infection-free steady state $E_{0}$ is globally asymptotically stable if $R_{0}\leq 1$ and becomes unstable if $R_{0}>1$.*


**Proof.** Let u=(S,L,I,V,W) be a solution of (Equation 1) and construct a Lyapunov functional as follows
$$\begin{array}{ccc} H_{0}\left(u\right) & = & S_{0}\Phi \left(\frac{S}{S_{0}}\right)+L+\frac{\mu_{2}+\delta +\rho }{\delta }I+\frac{\beta_{1}S_{0}}{\mu_{4}}V+\frac{p\beta_{1}S_{0}}{r\mu_{4}}W+\frac{\rho {\left(S-S_{0}+L\right)}^{2}}{2S_{0}\left(\mu_{1}+\mu_{2}+\delta \right)}, \end{array}$$
where Φ(x)=x−1−lnx, for x>0. By a simple computation, we have
$$\begin{array}{ccc} \frac{dH_{0}}{dt} & = & -\mu_{1}\left(\frac{1}{S}+\frac{\rho }{S_{0}\left(\mu_{1}+\mu_{2}+\delta \right)}+\frac{\rho L}{SS_{0}}\right){\left(S-S_{0}\right)}^{2}-\frac{\rho \left(\mu_{2}+\delta \right)L^{2}}{S_{0}\left(\mu_{1}+\mu_{2}+\delta \right)}\\ & & -\frac{q_{1}\beta_{1}S_{0}}{1+q_{1}W}VW-\frac{q_{2}\beta_{2}S_{0}}{1+q_{2}W}IW+\frac{\mu_{3}\left(\mu_{2}+\delta +\rho \right)}{\delta }I\left(R_{0}-1\right)\\ & & -\frac{p\mu_{5}\beta_{1}S_{0}}{r\mu_{4}}W. \end{array}$$
Hence, $\frac{dH_{0}}{dt}\leq 0$ for $R_{0}\leq 1$ and with equality if and only if $S=S_{0}$, L=0, I=0, V=0 and W=0. It follows from LaSalle’s invariance principle [34] that $E_{0}$ is globally asymptotically stable if $R_{0}\leq 1$.When $R_{0}>1$, the characteristic equation of model (Equation 1) at $E_{0}$ is given by
(10)$$\left(\mu_{1}+\lambda \right)\left(\mu_{5}+\lambda \right)P_{0}\left(\lambda \right)=0,$$
where
$$\begin{array}{ccc} P_{0}\left(\lambda \right) & = & \lambda^{3}+\left(\mu_{2}+\mu_{3}+\mu_{4}+\delta +\rho \right)\lambda^{2}\\ & & +(\mu_{3}\mu_{4}-\delta \beta_{2}S_{0}+\left(\mu_{3}+\mu_{4}\right)\left(\mu_{2}+\delta +\rho \right))\lambda \\ & & +\mu_{3}\mu_{4}\left(\mu_{2}+\delta +\rho \right)\left(1-R_{0}\right). \end{array}$$
We have $\lim_{\lambda \to +\infty }P_{0}\left(\lambda \right)=+\infty$ and $P_{0}\left(0\right)=\mu_{3}\mu_{4}\left(\mu_{2}+\delta +\rho \right)\left(1-R_{0}\right)<0$ if $R_{0}>1$. Therefore, the characteristic Equation (Equation 10) has at least one positive eigenvalue if $R_{0}>1$, which implies that $E_{0}$ is unstable when $R_{0}>1$. □

Next, we establish the asymptotic stability of the two infection steady states $E_{1}$ and $E_{2}$ by supposing that $R_{0}>1$ and the following further hypothesis
(*H*)$$\begin{array}{cc} \; & q_{1}\left(W-W_{i}\right)\left(\frac{1+q_{1}W}{1+q_{1}W_{i}}-\frac{V}{V_{i}}\right)\leq 0,\\ \; & q_{2}\left(W-W_{i}\right)\left(\frac{1+q_{2}W}{1+q_{2}W_{i}}-\frac{I}{I_{i}}\right)\leq 0, \end{array}$$
where $I_{i}$, $V_{i}$ and $W_{i}$ are productively infected cells, viruses and antibodies components of the infection equilibrium $E_{i}$, for i=1,2.

**Theorem** **4.**
*Assume that (Equation 11) is satisfied for $E_{1}$. Then the infection steady state without humoral immunity $E_{1}$ is globally asymptotically stable if $R_{1}^{W}\leq 1<R_{0}\leq 1+\frac{\mu_{2}+\delta }{\rho }$ and unstable if $R_{1}^{W}>1$.*


**Proof.** The characteristic equation at $E_{1}$ is given by
(11)$$\left(rV_{1}-\mu_{5}-\lambda \right)P_{1}\left(\lambda \right)=0,$$
where
$$P_{1}\left(\lambda \right)=\left|\begin{array}{cccc} -\mu_{1}-\beta_{1}V_{1}-\beta_{2}I_{1}-\lambda & \rho & -\beta_{2}S_{1} & -\beta_{1}S_{1}\\ \beta_{1}V_{1}+\beta_{2}I_{1} & -(\mu_{2}+\delta +\rho )-\lambda & \beta_{2}S_{1} & \beta_{1}S_{1}\\ 0 & \delta & -\mu_{3}-\lambda & 0\\ 0 & 0 & k & -\mu_{4}-\lambda \end{array}\right|.$$
Hence, $\lambda_{1}=rV_{1}-\mu_{5}$ is a root of the characteristic Equation (Equation 12). Since $R_{1}^{W}=\frac{rV_{1}}{\mu_{5}}>1$, we have $\lambda_{1}>0$. Then $E_{1}$ is unstable if $R_{1}^{W}>1$.For the global stability, we consider the following Lyapunov functional
$$\begin{array}{ccc} H_{1}\left(u\right) & = & S_{1}\Phi \left(\frac{S}{S_{1}}\right)+L_{1}\Phi \left(\frac{L}{L_{1}}\right)+\frac{\mu_{2}+\delta +\rho }{\delta }I_{1}\Phi \left(\frac{I}{I_{1}}\right)+\frac{\beta_{1}S_{1}V_{1}}{k\left(1-\epsilon \right)I_{1}}V_{1}\Phi \left(\frac{V}{V_{1}}\right)\\ & & +\frac{p\beta_{1}S_{1}V_{1}}{rk\left(1-\epsilon \right)I_{1}}W+\frac{\rho {\left(S-S_{1}+L-L_{1}\right)}^{2}}{2S_{1}\left(\mu_{1}+\mu_{2}+\mu_{3}\right)}. \end{array}$$
Then
$$\begin{array}{ccc} \frac{dH_{1}}{dt} & = & \left(1-\frac{S_{1}}{S}\right)\left(\sigma -\mu_{1}S-\frac{\beta_{1}SV}{1+q_{1}W}-\frac{\beta_{2}SI}{1+q_{2}W}+\rho L\right)\\ & & +\left(1-\frac{L_{1}}{L}\right)\left(\frac{\beta_{1}SV}{1+q_{1}W}+\frac{\beta_{2}SI}{1+q_{2}W}-\left(\mu_{2}+\delta +\rho \right)L\right)\\ & & +\frac{\mu_{2}+\delta +\rho }{\delta }\left(1-\frac{I_{1}}{I}\right)\left(\delta L-\mu_{3}I\right)\\ & & +\frac{\beta_{1}S_{1}V_{1}}{k\left(1-\epsilon \right)I_{1}}\left(1-\frac{V_{1}}{V}\right)\left(k\left(1-\epsilon \right)I-\mu_{4}V-pVW\right)\\ & & +\frac{p\beta_{1}S_{1}V_{1}}{rk\left(1-\epsilon \right)I_{1}}\left(rVW-\mu_{5}W\right)\\ & & +\frac{\rho }{\left(\mu_{1}+\mu_{2}+\mu_{3}\right)S_{1}}\left(S-S_{1}+L-L_{1}\right)\left(\sigma -\mu_{1}S-\left(\mu_{2}+\delta \right)L\right). \end{array}$$
Since $\sigma =\mu_{1}S_{1}+\beta_{1}S_{1}V_{1}+\beta_{2}S_{1}I_{1}-\rho L_{1}=\mu_{1}S_{1}+\left(\mu_{2}+\delta \right)L_{1}$, $\delta L_{1}=\mu_{3}I_{1}$ and $k\left(1-\epsilon \right)I_{1}=\mu_{4}V_{1}$, we get
$$\begin{array}{ccc} \frac{dH_{1}}{dt} & = & -\frac{1}{SS_{1}}\left(\mu_{1}S_{1}-\rho L_{1}+\frac{\rho \mu_{1}S}{\mu_{1}+\mu_{2}+\mu_{3}}+\rho L\right){\left(S-S_{1}\right)}^{2}\\ & & -\frac{\rho \left(\mu_{2}+\delta \right){\left(L-L_{1}\right)}^{2}}{\left(\mu_{1}+\mu_{2}+\mu_{3}\right)S_{1}}+\frac{p\mu_{5}\beta_{1}S_{1}}{r\mu_{4}}\left(R_{1}^{W}-1\right)W\\ & & +\beta_{1}S_{1}V_{1}\left(4-\frac{S_{1}}{S}+\frac{V}{\left(1+q_{1}W\right)V_{1}}-\frac{I_{1}L}{IL_{1}}-\frac{SVL_{1}}{\left(1+q_{1}W\right)LS_{1}V_{1}}-\frac{V}{V_{1}}-\frac{V_{1}I}{VI_{1}}\right)\\ & & +\beta_{2}S_{1}I_{1}\left(3-\frac{S_{1}}{S}+\frac{I}{\left(1+q_{2}W\right)I_{1}}-\frac{SIL_{1}}{\left(1+q_{2}W\right)S_{1}I_{1}L}-\frac{I}{I_{1}}-\frac{I_{1}L}{IL_{1}}\right). \end{array}$$
Thus,
$$\begin{array}{ccc} \frac{dH_{1}}{dt} & = & -\frac{1}{SS_{1}}\left(\mu_{1}S_{1}-\rho L_{1}+\frac{\rho \mu_{1}S}{\mu_{1}+\mu_{2}+\mu_{3}}+\rho L\right){\left(S-S_{1}\right)}^{2}\\ & & -\frac{\rho \left(\mu_{2}+\delta \right){\left(L-L_{1}\right)}^{2}}{\left(\mu_{1}+\mu_{2}+\mu_{3}\right)S_{1}}+\frac{p\mu_{5}\beta_{1}S_{1}}{r\mu_{4}}\left(R_{1}^{W}-1\right)W\\ & & +\beta_{1}S_{1}V_{1}\left(-1-\frac{V}{V_{1}}+\frac{V}{\left(1+q_{1}W\right)V_{1}}+\left(1+q_{1}W\right)\right)\\ & & +\beta_{2}S_{1}I_{1}\left(-1-\frac{I}{I_{1}}+\frac{I}{\left(1+q_{2}W\right)I_{1}}+\left(1+q_{2}W\right)\right)\\ & & +\beta_{1}S_{1}V_{1}\left(5-\frac{S_{1}}{S}-\frac{I_{1}L}{IL_{1}}-\frac{SVL_{1}}{\left(1+q_{1}W\right)S_{1}V_{1}L}-\frac{IV_{1}}{I_{1}V}-\left(1+q_{1}W\right)\right)\\ & & +\beta_{2}S_{1}I_{1}\left(4-\frac{S_{1}}{S}-\frac{I_{1}L}{IL_{1}}-\frac{SIL_{1}}{\left(1+q_{2}W\right)S_{1}I_{1}L}-\left(1+q_{2}W\right)\right). \end{array}$$
Then
$$\begin{array}{ccc} \frac{dH_{1}}{dt} & = & -\frac{1}{SS_{1}}\left(\mu_{1}S_{1}-\rho L_{1}+\frac{\rho \mu_{1}S}{\mu_{1}+\mu_{2}+\mu_{3}}+\rho L\right){\left(S-S_{1}\right)}^{2}\\ & & -\frac{\rho \left(\mu_{2}+\delta \right){\left(L-L_{1}\right)}^{2}}{\left(\mu_{1}+\mu_{2}+\mu_{3}\right)S_{1}}+\frac{p\mu_{5}\beta_{1}S_{1}}{r\mu_{4}}\left(R_{1}^{W}-1\right)W\\ & & +\beta_{1}S_{1}V_{1}\left(-1-\frac{V}{V_{1}}+\frac{V}{\left(1+q_{1}W\right)V_{1}}+\left(1+q_{1}W\right)\right)\\ & & +\beta_{2}S_{1}I_{1}\left(-1-\frac{I}{I_{1}}+\frac{I}{\left(1+q_{2}W\right)I_{1}}+\left(1+q_{2}W\right)\right)-\beta_{1}S_{1}V_{1}[\Phi \left(\frac{S_{1}}{S}\right)\\ & & +\Phi \left(\frac{I_{1}L}{IL_{1}}\right)+\Phi \left(\frac{SVL_{1}}{\left(1+q_{1}W\right)S_{1}V_{1}L}\right)+\Phi \left(\frac{IV_{1}}{I_{1}V}\right)+\Phi \left(1+q_{1}W\right)]\\ & & -\beta_{2}S_{1}I_{1}\left[\Phi \left(\frac{S_{1}}{S}\right)+\Phi \left(\frac{I_{1}L}{IL_{1}}\right)+\Phi \left(\frac{SIL_{1}}{\left(1+q_{2}W\right)S_{1}I_{1}L}\right)+\Phi \left(1+q_{2}W\right)\right]. \end{array}$$
According to (*Equation 11*), we obtain
(12)$$\begin{array}{cc} \; & -1-\frac{V}{V_{i}}+\frac{(1+q_{1}W_{i})V}{\left(1+q_{1}W\right)V_{i}}+\frac{1+q_{1}W}{1+q_{1}W_{i}}=\frac{q_{1}\left(W-W_{i}\right)}{1+q_{1}W}\left(\frac{1+q_{1}W}{1+q_{1}W_{i}}-\frac{V}{V_{i}}\right)\leq 0,\\ \; & -1-\frac{I}{I_{i}}+\frac{(1+q_{2}W_{i})I}{\left(1+q_{2}W\right)I_{i}}+\frac{1+q_{2}W}{1+q_{2}W_{i}}=\frac{q_{2}\left(W-W_{i}\right)}{1+q_{2}W}\left(\frac{1+q_{2}W}{1+q_{2}W_{i}}-\frac{I}{I_{i}}\right)\leq 0. \end{array}$$
If $R_{1}^{W}\leq 1$ and $\rho L_{1}\leq \mu_{1}S_{1}$, then $\frac{dH_{1}}{dt}\leq 0$ with equality if and only if $S=S_{1}$, $L=L_{1}$, $I=I_{1}$, $V=V_{1}$ and W=0. Further, the condition $\rho L_{1}\leq \mu_{1}S_{1}$ is equivalent to $R_{0}\leq 1+\frac{\mu_{2}+\delta }{\rho }$. Therefore, $E_{1}$ is globally asymptotically stable if $R_{1}^{W}\leq 1<R_{0}\leq 1+\frac{\mu_{2}+\delta }{\rho }$. □

**Remark** **1.**
*Since $\lim_{\rho \to 0^{+}}\frac{\mu_{2}+\delta }{\rho }=+\infty$ and $\lim_{\delta \to +\infty }\frac{\mu_{2}+\delta }{\rho }=+\infty$, we deduce that*
 **(i)** 
*$E_{1}$ is globally asymptotically stable if $R_{1}^{W}\leq 1<R_{0}$ and ρ=0.*
 **(ii)** 
*$E_{1}$ is globally asymptotically stable if $R_{1}^{W}\leq 1<R_{0}$ and δ is sufficiently large.*



**Theorem** **5.**
*Assume that (Equation 11) is satisfied for $E_{2}$. If $R_{1}^{W}>1$ and $\mu_{1}S_{2}-\rho L_{2}\geq 0$, then the infection steady state with humoral immunity $E_{2}$ is globally asymptotically stable.*


**Proof.** Consider the following Lyapunov functional
$$\begin{array}{ccc} H_{2}\left(u\right) & = & S_{2}\Phi \left(\frac{S}{S_{2}}\right)+L_{2}\Phi \left(\frac{L}{L_{2}}\right)+\frac{\mu_{2}+\delta +\rho }{\delta }I_{2}\Phi \left(\frac{I}{I_{2}}\right)\\ & & +\frac{\beta_{1}S_{2}V_{2}}{k\left(1-\epsilon \right)\left(1+q_{1}W_{2}\right)I_{2}}V_{2}\Phi \left(\frac{V}{V_{2}}\right)\\ & & +\frac{p\beta_{1}S_{2}V_{2}}{rk\left(1-\epsilon \right)\left(1+q_{1}W_{2}\right)I_{2}}W_{2}\Phi \left(\frac{W}{W_{2}}\right)+\frac{\rho {\left(S-S_{2}+L-L_{2}\right)}^{2}}{2S_{2}\left(\mu_{1}+\mu_{2}+\mu_{3}\right)}. \end{array}$$
By $\sigma =\mu_{1}S_{2}+\frac{\beta_{1}S_{2}V_{2}}{1+q_{1}W_{2}}+\frac{\beta_{2}S_{2}I_{2}}{1+q_{2}W_{2}}-\rho L_{2}=\mu_{1}S_{2}+\left(\mu_{2}+\delta \right)L_{2}$, $\delta L_{2}=\mu_{3}I_{2}$, $k\left(1-\epsilon \right)I_{2}=\mu_{4}V_{2}+pV_{2}W_{2}$, $rV_{2}=\mu_{5}$ and using the same technique of computation as in $H_{2}$, we obtain
$$\begin{array}{ccc} \frac{dH_{2}}{dt} & = & -\frac{1}{SS_{2}}\left(\mu_{1}S_{2}-\rho L_{2}+\frac{\rho \mu_{1}S}{\mu_{1}+\mu_{2}+\mu_{3}}+\rho L\right){\left(S-S_{2}\right)}^{2}\\ & & +\frac{-\rho (\mu_{2}+\delta )}{\left(\mu_{1}+\mu_{2}+\mu_{3}\right)S_{2}}{\left(L_{2}-L\right)}^{2}-\frac{\beta_{1}S_{2}V_{2}}{1+q_{1}W_{2}}[\Phi \left(\frac{I_{2}L}{L_{2}I}\right)+\Phi \left(\frac{S_{2}}{S}\right)\\ & & +\Phi \left(\frac{V_{2}I}{I_{2}V}\right)+\Phi \left(\frac{SL_{2}V\left(1+q_{1}W_{2}\right)}{S_{2}LV_{2}\left(1+q_{1}W\right)}\right)+\Phi \left(\frac{1+q_{1}W}{1+q_{1}W_{2}}\right)]\\ & & +\frac{\beta_{1}S_{2}V_{2}}{1+q_{1}W_{2}}\left(-1-\frac{V}{V_{2}}+\frac{V(1+q_{1}W_{2})}{V_{2}\left(1+q_{1}W\right)}+\frac{1+q_{1}W}{1+q_{1}W_{2}}\right)\\ & & -\frac{\beta_{2}S_{2}I_{2}}{1+q_{2}W_{2}}\left[\Phi \left(\frac{I_{2}L}{L_{2}I}\right)+\Phi \left(\frac{S_{2}}{S}\right)+\Phi \left(\frac{SL_{2}I\left(1+q_{2}W_{2}\right)}{S_{2}LI_{2}\left(1+q_{2}W\right)}\right)+\Phi \left(\frac{1+q_{2}W}{1+q_{2}W_{2}}\right)\right]\\ & & +\frac{\beta_{2}S_{2}I_{2}}{1+q_{2}W_{2}}\left(-1-\frac{I}{I_{2}}+\frac{I(1+q_{2}W_{2})}{I_{2}\left(1+q_{2}W\right)}+\frac{1+q_{2}W}{1+q_{2}W_{2}}\right). \end{array}$$
It follows from (Equation 13) and the condition $\mu_{1}S_{2}-\rho L_{2}\geq 0$ that $\frac{dH_{2}}{dt}\leq 0$ with equality if and only if $S=S_{2}$, $L=L_{2}$, $I=I_{2}$, $V=V_{2}$ and $W=W_{2}$. Hence, $E_{2}$ is globally asymptotically stable. □

## 4. Numerical Results

### 4.1. Parameters Estimation

Since the equilibrium point $E_{0}=\left(\frac{\sigma }{\mu_{1}},0,0,0,0\right)$ is the healthy state of our model, we deduce that $\frac{\sigma }{\mu_{1}}$ represents the total number of healthy pulmonary epithelial cells. According to [9], this number was estimated to be between $5.7757\times 10^{4}$ and $1.2\times 10^{7}$ cells/mL. It follows from [35] that $\mu_{1}=10^{-3}$ day${}^{-1}$. Thus, σ becomes between 57.757 and $1.2\times 10^{4}$ cells mL${}^{-1}$ day${}^{-1}$.

The rate to become productively infected cells, δ, was assumed to be 4 day${}^{-1}$ in [36,37], between 1 and 5 day${}^{-1}$ in [38]. Further, the parameter δ was estimated to be 7.88 day${}^{-1}$ in [39]. Then δ can be estimated between 1 and 7.88 day${}^{-1}$, which implies that the latently infected cells started producing virus about 3 to 24 h after they were infected.

The death rate of productively infected cells, $\mu_{3}$, was estimated to be between 1.5 and 5.2 day${}^{-1}$ in [36], 0.6 day${}^{-1}$ (95% confidence interval (CI) = 0.22–0.97) in [38], and around 1.7 day${}^{-1}$ in [37]. Hence, $\mu_{3}$ can be estimated between 0.6 and 5.2 day${}^{-1}$, corresponding to infected cells life-spans of 4 to 40 hours.

The virus-to-cell infection rate, $\beta_{1}$, was estimated to be between $4.8\times 10^{-8}$ and $4.5\times 10^{-5}$ mL virion${}^{-1}$ day${}^{-1}$ in [40], $3.2\times 10^{-8}$ mL virion${}^{-1}$ day${}^{-1}$ in [37], and $5\times 10^{-6}$ mL virion${}^{-1}$ day${}^{-1}$ in [36]. So, the parameter $\beta_{1}$ can be estimated between $3.2\times 10^{-8}$ and $4.5\times 10^{-5}$ mL virion${}^{-1}$ day${}^{-1}$.

The viral production, *k*, was estimated to be between 88 and 580 virions cell${}^{-1}$ day${}^{-1}$ in [9], 45.3 virions cell${}^{-1}$ day${}^{-1}$ in [37], and 22.71 virions cell${}^{-1}$ day${}^{-1}$ (95% CI = 0–59.94) in [38]. Hence, the parameter *k* can be estimated between 22.71 and 580 virions cell${}^{-1}$ day${}^{-1}$.

The clearance rate, $\mu_{4}$, was estimated to be 10 day${}^{-1}$ in [37], 2.44 and 15.12 day${}^{-1}$ in [9]. Further, Gonçalves et al. [38] assumed a viral clearance $\mu_{4}$ of 5 or 20 day${}^{-1}$ in order to characterize the viral load dynamics of 13 hospitalized patients in Singapore. Therefore, the parameter $\mu_{4}$ can be estimated between 2.44 and 20 day${}^{-1}$. The estimation of the other parameters are summarized in Table 1.

### 4.2. Sensitivity Analysis

Sensitivity analysis helps understand how changes in model parameters affect the dynamics of SARS-CoV-2 infection. Since the value of the basic reproduction number $R_{0}$ determines the eradication or the persistence of the virus in the human body, we so compute the sensitivity index of $R_{0}$ against model parameters. By using the explicit formula of $R_{0}$ given in (Equation 8), such index with respect a parameter *q* is calculated as follows
(13)$$\Gamma_{R_{0}}^{q}=\frac{q}{R_{0}}\frac{\partial R_{0}}{\partial q}.$$
Note that, by applying above Equation (Equation 14) and from the data in Table 1, we conclude that the most sensitive parameters to the basic reproduction number $R_{0}$ of the SARS-CoV-2 model are σ, $\beta_{1}$, $\beta_{2}$ and κ. Clearly, an increase of the value of any one of these parameters will increase the basic reproduction number. While, an increase of the value of other parameters ρ, $\mu_{1}$, $\mu_{2}$ and $\mu_{3}$ will decrease $R_{0}$ (Table 2). Then, we get the following results and illustrate it in Figure 2 and Figure 3.

### 4.3. Numerical Simulation

To illustrate our analytical results, we use the values of the parameters presented in Table 1 by choosing σ=60, $\mu_{1}=0.001$, $\mu_{2}=0.09$, δ=1.5, $\mu_{3}=0.75$, $\mu_{4}=15$, $\mu_{5}=0.3$, $\beta_{2}=1.2\times 10^{-8}$, $q_{1}=0.01$, $q_{2}=0.02$, p=0.5, ρ=0.01, ε=0.2, k=50 and the other parameters as *r* and $\beta_{1}$ are varied. Thus, the choice of the values of the two last parameters leads to the following scenarios:

**Scenario 1:** If we put $r=2.4\times 10^{-3}$ and $\beta_{1}=4.6\times 10^{-6}$, then $R_{0}=0.9209<1$. In this case, the dynamical behavior of model (Equation 1) converges to the infection-free equilibrium $E_{0}=\left(6\times 10^{4},0,0,0,0\right)$ from different initial values. This result is plotted by Figure 4.

**Scenario 2:** When we choose $\beta_{1}=1.3\times 10^{-5}$ and keeping the value of *r* in Scenario 1, we have $R_{0}=2.6009>1$ and $R_{1}^{W}=0.9910<1$. Figure 5 displays the dynamical behavior of model (Equation 1) approaches to $E_{1}=\left(2.3069\times 10^{4},23.2270,46.4541,12.3877,0\right)$ from different initial values.

**Scenario 3:** If $r=4\times 10^{-3}$ and $\beta_{1}=1.3\times 10^{-5}$ and keeping the value of $\beta_{1}$ in Scenario 2, we get $R_{1}^{W}=1.6517>1$ and the the dynamical behavior of model (Equation 1) approaches to globally asymptotically stable $E_{2}=\left(3.2263\times 10^{4},18.0990,36.2015,75.0259,8.6286\right)$ from various initial values. Figure 6 shows that result.

Finally, we study the impact of antiviral treatment on of the dynamics of SARS-CoV-2 infection. According to the explicit formula of the basic reproduction number $R_{0}$ presented in (Equation 8), we notice that $R_{0}$ is a decreasing function with respect to ϵ. From the numerical results (see, Figure 7), we notice the value of $R_{0}$ becomes less than 1 when the effectiveness of the antiviral treatment exceeds 70%. This biologically implies that SARS-CoV-2 infection will disappear from the human body when $R_{0}<1$.

## 5. Conclusions and Discussion

In this study, we have developed a new mathematical model that describes the dynamics of RNA viruses such as SARS-CoV-2 inside the human body. The developed model takes into account the two modes of transmission and both classes of infected cells that are latently infected cells and actively infected cells that produce virus particles. The cure of infected cells in latent period, both the lytic and non-lytic immune response are considered in our model. We first presented the result of the proposed model’s well-posedness in terms of non-negativity and boundedness of the solutions. We found two threshold parameters that completely determine the dynamics of this SARS-CoV-2 infection model while looking for biologically feasible equilibria. The first is the basic reproduction number, denoted by $R_{0}$ and the second is the reproduction number for humoral immunity, denoted by $R_{1}^{W}$. Our results indicate that the infection-free equilibrium $E_{0}$ is globally asymptotically stable when $R_{0}\leq 1$. This means that SARS-CoV-2 has been completely eradicated from human lungs. However, when $R_{0}>1$, $E_{0}$ becomes unstable and the virus persists in human lungs. Also, two scenarios appear depending on the value of the reproduction number for humoral immunity $R_{1}^{W}$. More precisely, the infection equilibrium without humoral immunity $E_{1}$ is globally asymptotically stable if $R_{1}^{W}\leq 1$. Whereas, when $R_{1}^{W}>1$, $E_{1}$ becomes unstable, and the infection-immunity equilibrium $E_{2}$ is globally asymptotically stable. Moreover, the sensitivity analysis showed that the evolution of the infection is influenced by the different parameters of the model. Finally, from the numerical simulations it is deduced that the use of the drugs makes the basic reproduction number $R_{0}$ of the patient less than or equal to 1, which automatically leads to the eradication of SARS-CoV-2 from the patient’s lungs. This shows the impact of treatment on the dynamics of infection by RNA viruses, in particular SARS-CoV-2. As well as, there are more results that we got it from sensitivity analysis as some parameters influence on the reproduction number. And then we get an accurate understanding of the behavior of the SARS-CoV-2 for example: $\sigma ,\beta_{1},\beta_{2}$ and *k*. Clearly, an increase of the value of any one of these parameters will increase $R_{0}$. While, an increase of the value of other parameters $\rho ,\mu_{1},\mu_{2}$ and $\mu_{3}$ will decrease $R_{0}$ (see Figure 2 and Figure 3).

Immunologic memory is a major feature of adaptive immunity. It consists in the ability of the immune system to recognize antigens, to which it was formerly exposed, and triggers a more robust and more efficient response than the first exposure [41]. For that reason, it will be more interesting to investigate the effect of immunological memory on the dynamics of the developed model as presented in [8,42,43]. This will be the our future scope.

## Figures and Tables

**Figure 1 vaccines-11-00201-f001:**
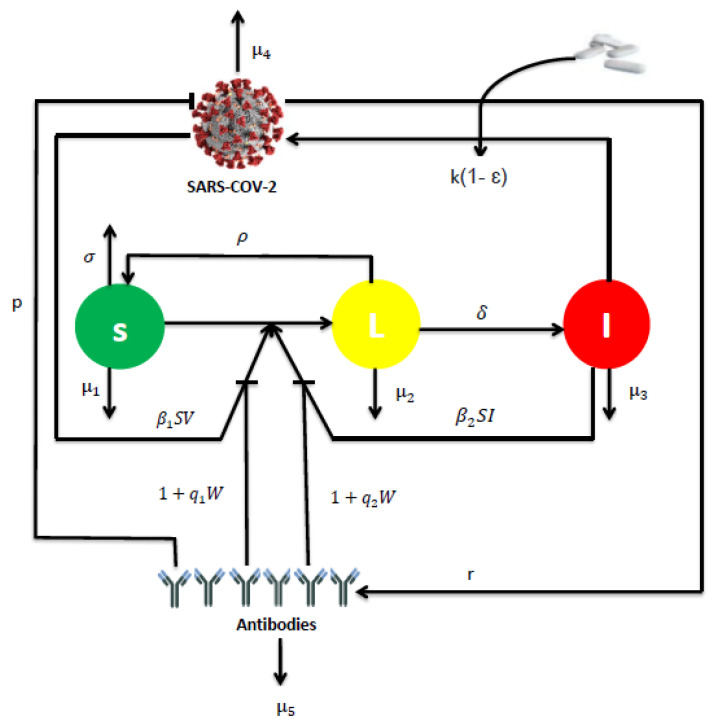
The flowchart representing the dynamics of model (Equation 1).

**Figure 2 vaccines-11-00201-f002:**
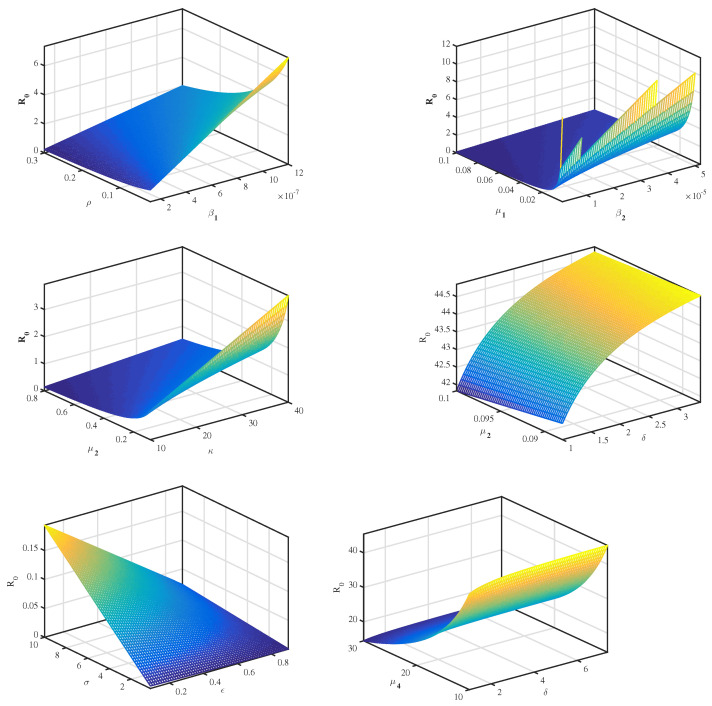
The relationship between $R_{0}$ and several parameters.

**Figure 3 vaccines-11-00201-f003:**
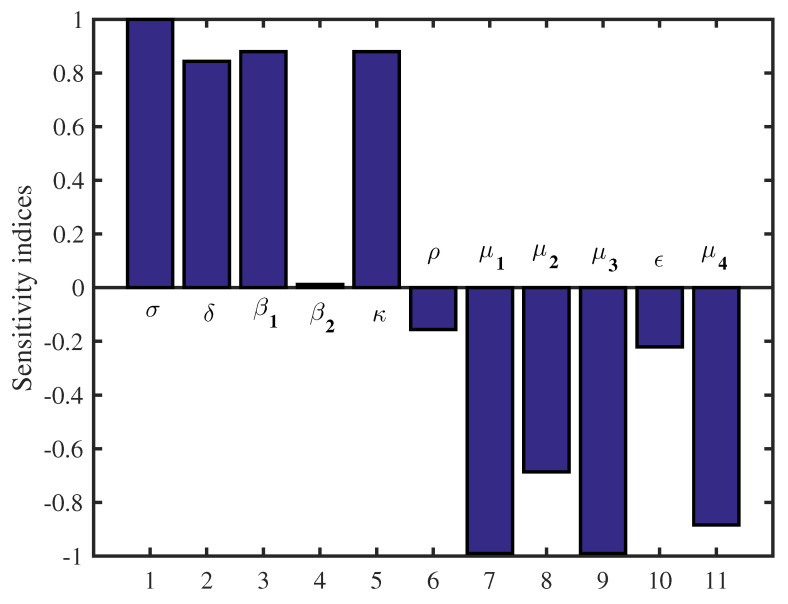
Sensitivity indices diagram for $R_{0}$.

**Figure 4 vaccines-11-00201-f004:**
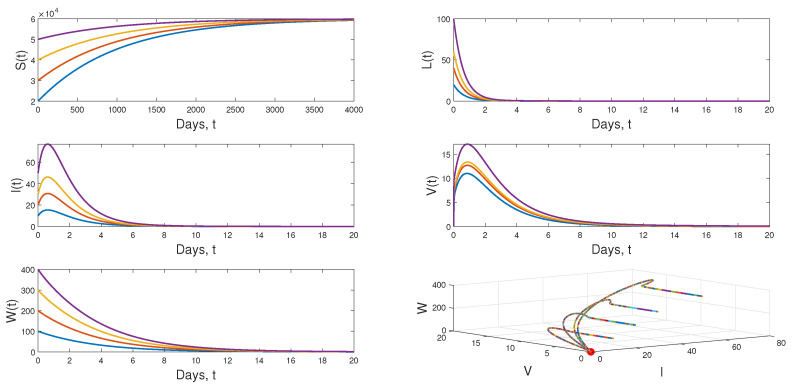
Dynamical behavior of model (Equation 1) when $R_{0}=0.9030<1$.

**Figure 5 vaccines-11-00201-f005:**
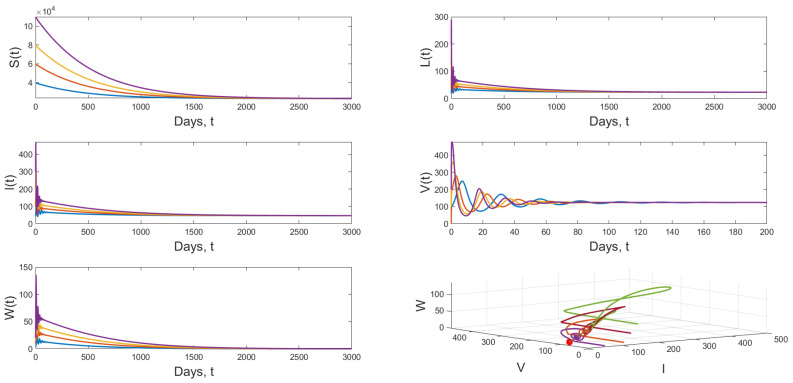
Dynamical behavior of model (Equation 1) when $R_{0}=2.6009>1$ and $R_{1}^{W}=0.9910<1$.

**Figure 6 vaccines-11-00201-f006:**
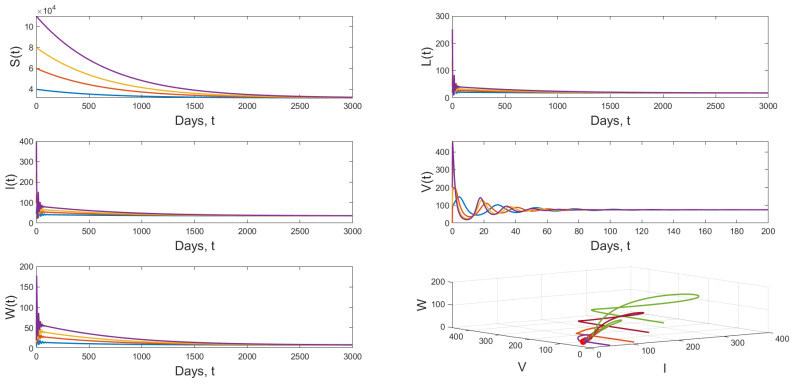
Dynamical behavior of model (Equation 1) when $R_{0}=2.6009>1$ and $E_{2}=\left(3.2263\times 10^{4},18.0990,36.2015,75.0259,8.6286\right)$.

**Figure 7 vaccines-11-00201-f007:**
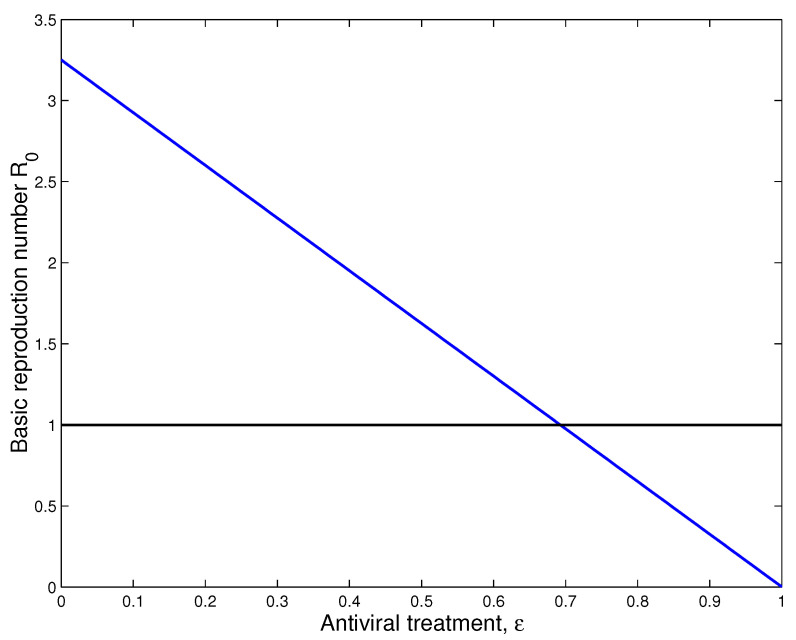
Impact of treatment on the dynamics of SARS-CoV-2 infection.

**Table 1 vaccines-11-00201-t001:** The 16 parameters of the model (Equation 1) with their values.

Parameter	Definition	Value	Source
σ	Epithelial cells	57.757–$1.2\times 10^{4}$	[9]
	production rate	cells mL${}^{-1}$ day${}^{-1}$	
$\mu_{1}$	Death rate of uninfected	$10^{-3}$ day${}^{-1}$	[35]
	epithelial cells		
$\beta_{1}$	Virus-to-cell infection rate	$3.2\times 10^{-8}$–$4.5\times 10^{-5}$	Estimated
		mL virion${}^{-1}$ day${}^{-1}$	
$\beta_{2}$	Cell-to-cell infection rate	0–1 mL cell${}^{-1}$ day${}^{-1}$	Assumed
$\mu_{2}$	Death rate of latently	0.08–0.59 day${}^{-1}$	Assumed
	infected epithelial cells		
δ	Rate to become productively	1–7.88 day${}^{-1}$	Estimated
	infected cells		
$\mu_{3}$	Death rate of productively	0.6–5.2 day${}^{-1}$	Estimated
	infected epithelial cells		
*k*	Virion production rate per	22.71–580	Estimated
	infected epithelial cell	virions cell${}^{-1}$ day${}^{-1}$	
$\mu_{4}$	Virus clearance rate	2.44–20 day${}^{-1}$	Estimated
*r*	Activation rate of	0–1 mL virion${}^{-1}$ day${}^{-1}$	Assumed
	antibodies		
$\mu_{5}$	Death rate of antibodies	0–1 day${}^{-1}$	Assumed
*p*	Neutralization rate of	0–1	Assumed
	virus by antibodies	mL molecules${}^{-1}$ day${}^{-1}$	
$q_{1}$	Non-lytic strength against	0–1 mL molecules${}^{-1}$	Assumed
	virus-to-cell infection		
$q_{2}$	Non-lytic strength against	0–1 mL molecules${}^{-1}$	Assumed
	cell-to-cell infection		
ρ	Cure rate of latently	0–1 day${}^{-1}$	Assumed
	infected cells		
ϵ	Effectiveness of	0–1	Assumed
	antiviral treatment		

**Table 2 vaccines-11-00201-t002:** Sensitivity of $R_{0}$ with respect to the parameters.

Parameter	Value	Sensitivity Index
σ	500	1
$\mu_{1}$	0.001	−1
$\beta_{1}$	0.0000011	0.88
$\beta_{2}$	0.00000012	0.0115
$\mu_{2}$	0.088	−0.687
δ	4.5	0.844
$\mu_{3}$	0.088	−1
*k*	88	0.88
$\mu_{4}$	10	−0.885
ρ	0.02	−0.156
ϵ	0.2	−0.2212

## Data Availability

Not applicable.

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
