# Peer review of "Mathematical Modeling and Analysis of the Dynamics of RNA Viruses in Presence of Immunity and Treatment: A Case Study of SARS-CoV-2"

_vaccines, 2023, doi:10.3390/vaccines11020201_

Round 1

Reviewer 1 Report

This paper develops a new mathematical model to better understand the evolutionary behavior of RNA viruses like SARS-CoV-2 inside the human body. The developed model takes into account two modes of transmission and both classes of infected cells that are latently infected cells and actively infected cells that produce virus particles. The cure of infected cells in latent period as well as the lytic and non-lytic immune response are considered into the model.

   The paper is well written, clear and contains very interesting results. The paper can be published after some minor revisions.

1.    The biological meanings of each parameter of your model should be checked.

2.    Write the conclusion with future scope related to fractional operators with stability analysis. So that it can be a better read for researchers working in the Mathematical fractional modeling. The authors can refer to the articles: https://doi.org/10.1140/epjs/s11734-022-00458-0, AIMS Mathematics 2023, Volume 8, Issue 1: 1656-1671. doi:10.3934/math.2023084.

3.    Compare the model presented by system (1) with some mathematical models for viral infection existing in the literature.

4.    Write the biological meaning of the threshold parameter R0.

5.    Unify the style of the references.

Author Response

Response to Reviewer 1
We thank the reviewer for the useful comments on our manuscript. A response to each of the comments is given below.
Comments: This paper develops a new mathematical model to better understand the evolutionary behavior of RNA viruses like SARS-CoV-2 inside the
human body. The developed model takes into account two modes of transmission and both classes of infected cells that are latently infected cells and
actively infected cells that produce virus particles. The cure of infected cells in
latent period as well as the lytic and non-lytic immune response are considered
into the model.
The paper is well written, clear and contains very interesting results. The
paper can be published after some minor revisions.
—Reply:
Thank you for your positive comments on the manuscript and encouraging
words on the quality of the manuscript.
1. The biological meanings of each parameter of your model should be checked.
Response: We checked it.
2. Write the conclusion with future scope related to fractional operators with
stability analysis. So that it can be a better read for researchers working
in the Mathematical fractional modeling. The authors can refer to the articles: https://doi.org/10.1140/epjs/s11734-022-00458-0, AIMS Mathematics 2023, Volume 8, Issue 1: 1656-1671. doi:10.3934/math.2023084.
Response: We added it, see the revised version of the manuscript.
3. Compare the model presented by system (1) with some mathematical models for viral infection existing in the literature.
Response: We compared it.
4. Write the biological meaning of the threshold parameter R0.
Response: We added it.
5. Unify the style of the references.
Response: We fixed it. Thank you very much for your interesting comments.

Reviewer 2 Report

In this study, the authors proposed and analyzed a new mathematical model describing the dynamics of RNA viruses in presence of humoral immunity and antiviral treatment.The article is logically correct and the results obtained are good and interesting.    Therefore, I recommend this article for publication after a minor revision.   1)  Include more motivations for your new model (1). 2) On page 1, remove "These authors contributed equally to this work". 3) Prove the existence of solutions of your model (1). 4) Check that all references are cited in the text.  

Author Response

Response to Reviewer 2
We thank the reviewer for the useful comments on our manuscript. A response to each of the comments is given below.
Comments: In this study, the authors proposed and analyzed a new mathematical model describing the dynamics of RNA viruses in presence of humoral
immunity and antiviral treatment.The article is logically correct and the results obtained are good and interesting. Therefore, I recommend this article
for publication after a minor revision.
—Reply:
Thank you for your positive comments on the manuscript and encouraging
words on the quality of the manuscript.
1. Include more motivations for your new model (1).
Response: We included it, see the revised version.
2. On page 1, remove ”These authors contributed equally to this work”.
Response: We removed it.
3. Prove the existence of solutions of your model (1).
Response: We proved it.
4. Check that all references are cited in the text.
Response: We checked it. Thank you very much.

Author Response

Response to Reviewer 3
We thank the reviewer for the useful comments on our manuscript. A response to each of the comments is given below.
Comments: I think the paper proposes a novel mathematical model of the dynamics of RNA viruses in presence of immunity and treatment: A case study
of SARS-CoV-2. Unfortunately, although the research content of this paper
is novel and logical, it still needs to be improved before it is accepted and
published.
—Reply:
Thank you for your positive comments on the manuscript and encouraging
words on the quality of the manuscript.
3
1. The description of the literature is not enough. It is suggested that the
authors explain where this model differs from previous studies clearly.
I think the authors should explain from the oldest to recent researches.
Please explain the innovation and breakthrough of this model in introduction more explicitly.
Response: We fixed it.
2. How the authors find R0? What is the method used? In addition, how
the authors find RW
1
? What is the difference between R0 and RW
1
in
mathematical and biological meanings?
Response: R0 can be calculated by applying the next generation matrix
approach. Here, this number is determined from the calculation of the
equilibrium points. R0 is the threshold parameter which allows the transition from free equilibrium to endemic equilibrium. After, we check that
the free equilibrium is stable if R0 < 1 and becomes unstable if R0 > 1
(see Theorem 3). R0 is the threshold parameter for viral infection, but
RW
1
is for humoral immunity. Also, the biological meanings R0 and RW
1
have been added, see the revised version.
3. How the authors construct Lyapunov functions H0(u), H1(u) and H1(u)?
Response: The construction of Lyapunov functions is based on the direct
Lyapunov method.
4. I think the authors add conclusion section to explain the main results in
this paper.
Response: We added it.
5. Please check the English language throughout the whole manuscript.
Response: We checked it. Thank you very much.

Reviewer 4 Report

The manuscript presents a mathematical model of SARS-CoV-2 within host infection that includes two transmission modes, humoral immunity, and antiviral treatment. There are some serious flaws with both the model and the manuscript:

1. In the introduction, the authors state "To date, studies have been unable to identify a sufficiently potent antiviral treatment for SARS-CoV-2 infection [3]." This is soon followed by "A recent study confirmed that Paxlovid is effective at 89% in high-risk patients [4]" How can there be no potent antiviral treatment, when an antiviral with 89% effectiveness exists?

2. There is really no discussion of what is known of the humoral antibody response in the introduction, even though the authors claim that the model is supposed to investigate this.

3. There is also little discussion of previous modeling efforts for SARS-CoV-2, some of which have included treatment and antibody responses.

4. It's not clear why humoral immunity reduces the cell-to-cell transmission rate. Last I checked, anitbodies only exist outside cells, so should not interfere with viruses tunneling from one cell to another.

5. It's also odd to have antibodies reduce the cell-free infection rate, what is the mechanism that causes this? The model already explicitly includes the effect of antibodies binding to free virus, so what is causing the reduction in infection rate?

6. What evidence is there that latently infected cells return back to susceptible cells during SARS-CoV-2 infection? The loss of cccDNA as a mechanism here doesn't make any sense because this is an RNA virus.

7. It is not at all clear what parameter values were used for the simulations. There is one set of parameters in Table 1. There is another set of parameters in Table 2 (without units). The text states that the parameters in Table 2 are used for simulation, but then goes on to list a third set of parameters (also without units) that does not agree with either Table 1 or Table 2.

8. There is no discussion or conclusion. What am I supposed to have learned from this exercise?

Author Response

Response to Reviewer 4
We thank the reviewer for the useful comments on our manuscript. A response to each of the comments is given below.
Comments: The manuscript presents a mathematical model of SARS-CoV-2
within host infection that includes two transmission modes, humoral immunity, and antiviral treatment.
—Reply:
Thank you for your positive comments on the manuscript and encouraging
words on the quality of the manuscript.
1. In the introduction, the authors state ”To date, studies have been unable to identify a sufficiently potent antiviral treatment for SARS-CoV-2
infection [3].” This is soon followed by ”A recent study confirmed that
Paxlovid is effective at 89% in high-risk patients [4]” How can there be
no potent antiviral treatment, when an antiviral with 89% effectiveness
exists?
Response: We fixed it.
2. There is really no discussion of what is known of the humoral antibody response in the introduction, even though the authors claim that the model
is supposed to investigate this.
Response: We added it, see the introduction section of the revised version.
3. There is also little discussion of previous modeling efforts for SARS-CoV2, some of which have included treatment and antibody responses.
Response: We fixed it.
4. It’s not clear why humoral immunity reduces the cell-to-cell transmission
rate. Last I checked, anitbodies only exist outside cells, so should not
interfere with viruses tunneling from one cell to another.
Response: We fixed it by adding the following, ”Accumulating data
shows that most viruses can spread efficiently through cell-to-cell transmission. This mode of infection is resistant to circulating antibodies.
This was reported for SARS-COV-2 as well. In an in vitro study it was
shown that SARS-COV-2 cell-free infection was almost completely inhibited by neutralizing monoclonal antibodies and COVID-19 convalescent
plasma, whereas cell-to-cell infection was substantially resistant to this
treatment [14]”.
5. It’s also odd to have antibodies reduce the cell-free infection rate, what is
the mechanism that causes this? The model already explicitly includes the
effect of antibodies binding to free virus, so what is causing the reduction
in infection rate?
Response: As we mentioned previously an in vitro study has shown that
cell-free infection is sensitive to circulating neutralizing antibodies.
6. What evidence is there that latently infected cells return back to susceptible cells during SARS-CoV-2 infection? The loss of cccDNA as a
mechanism here doesn’t make any sense because this is an RNA virus.
Response: We corrected it.
7. It is not at all clear what parameter values were used for the simulations.
There is one set of parameters in Table 1. There is another set of parameters in Table 2 (without units). The text states that the parameters
in Table 2 are used for simulation, but then goes on to list a third set of
parameters (also without units) that does not agree with either Table 1
or Table 2.
Response: We corrected it.
8. There is no discussion or conclusion. What am I supposed to have learned
from this exercise?
Response: We added a section for conclusion and discussion. Thank you
very much your interesting comments.

Round 2

Reviewer 1 Report

I am satisfied with the contents of the revised version. So, the manuscript can be processed to publication.

Author Response

Response to Reviewer 1
We thank the reviewer for the useful comments on our manuscript. A response to each of the comments is given below.
Comments: I am satisfied with the contents of the revised version. So, the
manuscript can be processed to publication.
—Reply:
Thank you very much for recommending our paper to the publication in the
Journal.

Reviewer 3 Report

I would like to thank the authors for their efforts. Checking the paper for plagiarism, I found it to be more than 30%. This should be reduced.

Author Response

Response to Reviewer 3
We thank the reviewer for the useful comments on our manuscript. A response to each of the comments is given below.
Comments: I would like to thank the authors for their efforts.
—Reply:
Thank you for your positive comments on the manuscript and encouraging
words on the quality of the manuscript.
1. Checking the paper for plagiarism, I found it to be more than 30%. This
should be reduced.
Response: We reduced it. Thank you very much.

Reviewer 4 Report

The authors have addressed some of my previous comments, but have not changed the model. Since the model seems to include mechanisms that are not justified for SARS-CoV-2, any predictions made by the model are not relevant for SARS-CoV-2. Specifically,

1. While the authors removed the reference to cccDNA as a mechanism for reversion of latent cells to susceptible cells, they have not provided evidence that SARS-CoV-2 does this. They cite the fact that HCV and HIV do this, but these are both chronic infections where the latent phase can last a fairly long time. Without actual evidence that SARS-CoV-2 latently infected cells revert back to susceptible, this process must be removed from the model.

2. The authors agree with me that antibodies do not reduce cell-to-cell transmission, so why is beta_2 reduced by the presence of antibodies? In fact, the effect is apparently stronger for cell-to-cell transmission than for cell-free transmission since q_2>q_1. The authors again have included a process for which there is no justification, making the model an incorrect representation of SARS-CoV-2.

3. Finally, there is still no explanation of how antibodies are reducing the infection rate. While it is true that the presence of antibodies slows down or stops an infection, this is because antibodies bind to and remove virus --- a mechanism already incorporated into the model. Now there are fewer free viral particles floating around, so there are fewer cells infected, but it is not because the probability of infection has gone down. The authors have not presented a biological mechanism that, separate from antibody/virus binding, reduces the infection rate.

Other issues:

1. While the introduction is somewhat improved, there are now many statements of fact that do not have references (all the details of when various antibodies appear are unreferenced).

2. There are still inconsistencies in the parameter values. For example, b_2 in Table 1 is between 0-1, but in Table 2 and line 231, it has a value of 1.2e-8. delta_2 in Table 2 is 4.5 on line 230, it is 1.5. mu_4 is 10 in Table 2 and 15 on line 230, and there are other inconsistencies.. 

Author Response

Response to Reviewer 4
We thank the reviewer for the useful comments on our manuscript. A response to each of the comments is given below.
Comments: The authors have addressed some of my previous comments, but
have not changed the model. Since the model seems to include mechanisms
that are not justified for SARS-CoV-2, any predictions made by the model are
not relevant for SARS-CoV-2.

—Reply:
Thank you for your positive comments on the manuscript and encouraging
words on the quality of the manuscript.
1. While the authors removed the reference to cccDNA as a mechanism
for reversion of latent cells to susceptible cells, they have not provided
evidence that SARS-CoV-2 does this. They cite the fact that HCV and
HIV do this, but these are both chronic infections where the latent phase
can last a fairly long time. Without actual evidence that SARS-CoV2 latently infected cells revert back to susceptible, this process must be
removed from the model.
Response: In this case, we have ρ = 0. Our model includes this particular
case. Also, we added this special case of our model, see Section 2 of the
revised version.
2. The authors agree with me that antibodies do not reduce cell-to-cell transmission, so why is β2 reduced by the presence of antibodies? In fact, the
effect is apparently stronger for cell-to-cell transmission than for cell-free
transmission since q2 > q1. The authors again have included a process
for which there is no justification, making the model an incorrect representation of SARS-CoV-2.
Response: When antibodies do not reduce cell-to-cell transmission, we
have q2 = 0. The particular case of our model was added, see Section 2
of the revised version. Furthermore, some biological references are added
in order to justify that antibodies inhibit the infection of susceptible cells
and contribute significantly to non-lytic antiviral activity.
3. Finally, there is still no explanation of how antibodies are reducing the
infection rate. While it is true that the presence of antibodies slows down
or stops an infection, this is because antibodies bind to and remove virus
— a mechanism already incorporated into the model. Now there are
fewer free viral particles floating around, so there are fewer cells infected,
but it is not because the probability of infection has gone down. The
authors have not presented a biological mechanism that, separate from
antibody/virus binding, reduces the infection rate.
Response: We fixed it by adding some biological references.
4. While the introduction is somewhat improved, there are now many statements of fact that do not have references (all the details of when various
antibodies appear are unreferenced).
Response: We fixed it.
5. There are still inconsistencies in the parameter values. For example, β2
in Table 1 is between 0-1, but in Table 2 and line 231, it has a value of1.2e − 8. delta2 in Table 2 is 4.5 on line 230, it is 1.5. mu4 is 10 in
Table 2 and 15 on line 230, and there are other inconsistencies.
Response: There are no inconsistencies in the parameter values. Indeed,
In table 1, we have β2 ∈ [0, 1]. Hence, β2 = 1.2 × 10−8 ∈ [0, 1]. Similarly
for the others parameters.

Round 3

Reviewer 4 Report

It's pretty clear the authors do not want to modify the model. If they insist on keeping the latent reversion, then they have a model of HCV or HIV, not SARS-CoV-2. I realize that they can claim that rho=0 (no reversion) is a special case of their model, but that is not the results they are presenting here. They simply cannot claim that they are modeling SARS-CoV-2 if they keep this mechanism in the model.

As for the antibodies reducing infection rate, I still don't see a justification for that. The citation for "evidence" is a chapter in a modeling book. That particular chapter deals with CTLs, not antibodies, so is not relevant. Not to mention that just because somebody includes something in a model doesn't mean it actually happens in real life (take the model in this paper, for example). The comment that this represents the "non-lytic" activity of antibodies is also confusing since antibody binding to virus is non-lytic (lysis is a cellular process and antibodies don't directly affect cells in this way).

The paper has a model that incorporates two biological mechanisms that do not occur for the infection that the authors claim to be modeling, This is scientifically incorrect. They can either modify the model so that it correctly captures the behavior of the specified disease or find a disease that actually has these particular mechanisms and say they are modeling that.

Author Response

Dear Editor,
Thank you for supervising the review of our paper entitled ” Mathematical
modeling and analysis of the dynamics of RNA viruses in presence of immunity
and treatment: A case study of SARS-CoV-2”. We would like to thank the
anonymous reviewers for their valuable comments and suggestions. We have
revised the paper carefully in accordance with these comments and suggestions.
Best regards
Response to Reviewer 4
We thank the reviewer for the useful comments on our manuscript. A response
to each of the comments is given below.

1. It’s pretty clear the authors do not want to modify the model. If they
insist on keeping the latent reversion, then they have a model of HCV
or HIV, not SARS-CoV-2. I realize that they can claim that rho=0 (no
reversion) is a special case of their model, but that is not the results they
are presenting here. They simply cannot claim that they are modeling
SARS-CoV-2 if they keep this mechanism in the model.

Response: In general, we mentioned on page 2, line 65, that our model
describes the dynamics of RNA viruses. In particular, the choice of the
parameters values of this model for a given RNA virus can describe the
dynamical behavior of a specific disease. In addition, we added a recent
reference for SARS-CoV-2 that the cure of infected epithelial cells was
considered. Therefore, our model can describes the dynamics of RNA
viruses such as SARS-CoV-2, HCV and HIV.

2. As for the antibodies reducing infection rate, I still don’t see a justification for that. The citation for ”evidence” is a chapter in a modeling book.
That particular chapter deals with CTLs, not antibodies, so is not relevant. Not to mention that just because somebody includes something in a
model doesn’t mean it actually happens in real life (take the model in this
paper, for example). The comment that this represents the ”non-lytic”
activity of antibodies is also confusing since antibody binding to virus is
non-lytic (lysis is a cellular process and antibodies don’t directly affect
cells in this way).

Response: We have included two references in order to justify that antibodies inhibit the infection of susceptible cells and contribute significantly to non-lytic antiviral activity. The first of these references waspublished in ”Trends in Immunology” which plays an essential role in monitoring advances in the various fields of immunology.

3. The paper has a model that incorporates two biological mechanisms that
do not occur for the infection that the authors claim to be modeling, This
is scientifically incorrect. They can either modify the model so that it
correctly captures the behavior of the specified disease or find a disease
that actually has these particular mechanisms and say they are modeling
that.

Response: Our model describes the dynamics of RNA viruses. The analytical results obtained in Section 3 are illustrated by numerical simulations in Section 4 based on parameters estimation of SARS-CoV-2 given
in Subsection 4.1. Furthermore, all biological mechanisms included into
our model are justified.
